# Maternal, fetal, and perinatal outcomes among pregnant women admitted to an Ebola treatment center in the Democratic Republic of Congo, 2018–2020

**David Philpott**[1]*, **Neil Rupani**[2], **Monique Gainey**[3], **Eta N. Mbong**[4], **Prince Imani Musimwa**[5], **Shiromi M. Perera**[6], **Razia Laghari**[4], **Mija Ververs**[7], **Adam C. Levine**[8]

1 Johns Hopkins Children's Center, Baltimore, Maryland, United States of America, 2 Brown University, Providence, Rhode Island, United States of America, 3 Department of Emergency Medicine, Rhode Island Hospital, Providence, Rhode Island, United States of America, 4 International Medical Corps, Goma, Democratic Republic of Congo, 5 Department of Gynecology and Obstetrics, University of Goma, Goma, Democratic Republic of Congo, 6 International Medical Corps, Washington, District of Colombia, United States of America, 7 Center for Humanitarian Health, Johns Hopkins Bloomberg School of Public Health, Baltimore, Maryland, United States of America, 8 Department of Emergency Medicine, Brown University, Providence, Rhode Island, United States of America

* dcephilpott@gmail.com

## Abstract

### Objective

This study aims to investigate maternal, fetal, and perinatal outcomes during the 2018–2020 Ebola outbreak in Democratic Republic of Congo (DRC).

### Methods

Mortality between pregnant and non-pregnant women of reproductive age admitted to DRC's Mangina Ebola treatment center (ETC) were compared using propensity score matching. Propensity scores were calculated using age, initial Ebola viral load, Ebola vaccination status, and investigational therapeutic. Additionally, fetal and perinatal outcomes of pregnancies were also described.

### Results

Twenty-seven pregnant women were admitted to the Mangina ETC during December 2018—January 2020 among 162 women of childbearing age. We found no evidence of increase mortality among pregnant women compared to non-pregnant women (relative risk:1.0, 95%CI: 0.58–1.72). Among surviving mothers, pregnancy outcomes were poor with at least 58% (11/19) experiencing loss of pregnancy while 16% (3/19) were discharged with viable pregnancy. Two mothers with viable pregnancies were vaccinated, and all received investigational therapeutics. Two live births occurred, with one infant surviving after the infant and mother received an investigational post-exposure prophylaxis and Ebola therapeutic respectively.

**Data Availability Statement:** We obtained minimal data from a third party, the International Medical Corps (IMC) and data are not owned the authors and therefore cannot be shared. The data

underlying the results presented in the study are freely available from the International Medical Corps Research Review Committee. We confirm authors did not have any special access or request privileges that others would not have. The link to access the data can be found on the IMC website Ebola Response page at: http://internationalmedicalcorps.org/document.doc?id=800.

**Funding:** The author(s) received no specific funding for this work.

**Competing interests:** The authors have declared that no competing interests exist.

## Conclusions

Pregnancy was not associated with increased mortality among women with EVD in the Mangina ETC. Fetal and perinatal outcomes remained poor in pregnancies complicated by EVD, though novel therapeutics may have potential for improving these outcomes.

## Introduction

On August 1, 2018, the Democratic Republic of the Congo (DRC)'s Ministry of Health declared an Ebola virus disease (EVD) outbreak in the provinces of North Kivu and Ituri. This tenth EVD outbreak resulted in 3,470 cases and 2,287 deaths and was the largest in DRC's history [1–3].

Despite these outbreaks, knowledge of EVD in pregnancy is limited. Few studies report maternal and fetal outcomes of pregnant women infected with Ebola virus (EBOV). This gap exists partly because surveillance systems have not consistently recorded pregnancy status [4]. Historical studies among pregnant women with EVD reported mortality as high as 90% and poor perinatal outcomes [5, 6]. However, more recent data suggest mortality among pregnant and non-pregnant women with EVD is similar [6, 7].

Additionally, existing literature reports poor fetal outcomes among women with EVD including miscarriage, stillbirth, or death during the neonatal period [7–12]. One study during the 2014–2016 West Africa outbreak found 83% of neonates born to mothers with EVD died [7]. Likewise, while a study describing maternal and fetal outcomes in the 2014–2016 West Africa outbreak found no evidence of increased mortality secondary to EVD among pregnant women, all pregnancies among women who survived EVD resulted in fetal demise, termination, or infant death within the first two weeks of life [12]. Recently, a case series reported two infants born to women with EVD while in an ETC who survived after both mother and infant received monoclonal antibody therapy [13].

Obstetric care guidelines for women with EVD historically suggested a case-specific approach that balances risks and benefits of pregnancy related interventions for the mother with the risk of exposure to healthcare workers while assuming poor fetal outcome [12, 14]. In February 2020, the World Health Organization (WHO) released guidance on the management of pregnant and breastfeeding women who have EVD [15]. These guidelines recommend offering investigational therapies MAb114 and REGN-EB3 to pregnant women, cautioned against induction of labor for pregnant women with EVD, avoidance of invasive procedures for fetal indications, infection control measures for neonates delivered to mothers with history of EVD during pregnancy, and that women with history of EVD should give birth at an Ebola Treatment Center [15].

No study has reported comprehensive maternal or fetal outcomes from the tenth DRC Ebola outbreak. Our study aimed to compare EVD outcomes and symptoms between pregnant and non-pregnant women with EVD managed at an ETC in eastern DRC during the tenth Ebola outbreak. Additionally, we sought to describe fetal outcomes of pregnant women with EVD.

## Methods

Our study utilized data collected for a retrospective cohort study at the International Medical Corps (IMC) Mangina Ebola Treatment Center (ETC) during the tenth Ebola Virus Disease (EVD) outbreak in the Democratic Republic of the Congo (DRC). All patients who presented

to the Mangina ETC between December 2018 and January 2020 that met the WHO case definition for EVD were eligible for inclusion in the overall cohort [16]. For this study, we only included female patients of child-bearing age, defined as aged 15–50 years with laboratory confirmed EVD.

Patients were screened by trained clinicians to ensure they met the case definition for suspected EVD based on WHO guidelines [17]. Patients who met the case definition without laboratory confirmation were admitted to the suspect ward, where they received initial EVD testing. Patients with a positive test were moved to the confirmed ward, while those who tested negative remained in the ETC for 72 hours to ensure they remained without EVD symptoms and had a second negative test.

Patients were tested for EBOV by reverse transcriptase polymerase chain reaction (RT-PCR) using the Cepheid GeneXpert Ebola assay which determined the plasma cycle threshold (Ct) of EBOV [18]. A Ct greater than 40 was considered negative. Daily testing was conducted, and patients were discharged after two consecutive negative tests. The Mangina ETC served as a PALM Trial site with patients randomized to investigational therapeutic agents for EVD, including monoclonal antibody treatments (MAb114, REGN-EB3, and ZMapp) and the antiviral Remdesivir [19]. This trial allowed neonates born in ETCs to mothers with EVD to receive an investigational therapeutic regardless of EVOD test result at birth [19].

Deidentified data for the cohort included two sources: First was a line-list of patients admitted to the Mangina ETC. Second, a database was created using standardized forms in patient charts. ETC staff uploaded scanned charts to IMC's server and research staff transcribed this information into an Excel database. This database added vitals, exams, laboratory results, and medications. After data entry was complete, staff audited a random sample of 62 (15%) records by reentering data from original charts into a second database. There was 97.3% similarity between the two databases. For further quality assurance, the line list and EVD positive database were reconciled across 145 common variables.

Our exposure, pregnancy, was identified by self-report, clinician observation, or positive pregnancy test. For quality assurance, we reviewed records of patients who received misoprostol or oxytocin but were not documented as pregnant and if they were found to be pregnant added them to the database. Additionally, we reviewed all pregnant patients to ensure completeness of pregnancy related data including last menstrual period (LMP), trimester of pregnancy on admission, and obstetric complications such as post-partum hemorrhage and unexpected uterine contractions. Patients with missing pregnancy status were classified as non-pregnant given the proportion of population documented as pregnant was similar to that expected by authors and field staff who cared patients in the Mangina ETC.

Our primary maternal outcome, maternal death, was documented in the EVD positive database. Patients transferred to another facility were coded as having survived. Patients missing outcome data were coded as survived if their last documented RT-PCR was negative (Cycle threshold (Ct)>40). After reviewing available data, we classified pregnancy outcomes into groups including: spontaneous abortion, fetal demise with subsequent induction of delivery, live birth, and discharge or transfer with a viable pregnancy. We also reviewed available data for any live births in the ETC.

We used descriptive statistics to compare pregnant and non-pregnant patients in the cohort by age, self-reported Ebola vaccination status, presenting signs and symptoms at triage, days since symptom onset, initial viral load (Ct), and investigational therapeutic agent.

We used a propensity score matching approach to compare mortality between EVD pregnant and non-pregnant women while controlling for confounders. We attempted multiple matching techniques to ensure adequate balance between groups including genetic and nearest

neighbor matching [20, 21]. We selected variables for inclusion on the basis of prior analyses of risks for mortality in EVD patients [22, 23]. These variables were patient age, Ct-value on admission, days since symptom onset, vaccination status, and experimental therapeutic received. Patients were exactly matched on vaccination status given its likely association with mortality [24]. Propensity scores were estimated by fitting a logistic regression model. We utilized a common support interval with patients not falling within the region of common support excluded to prevent overextrapolation. In addition, we used a caliper with distance equal to 0.25 of the standard deviation of the propensity score. Balance on variables of interest before and after propensity score matching was assessed using standardized mean differences and Love plots. The acceptable standardized difference between groups was set at 0.1, consistent with established guidance [25]. We then calculated the relative risk comparing the two groups using log-binomial regression using cluster-robust standard errors and pair membership as the clustering variable. All analyses were conducted in R version 4.1 (R Foundation).

The Institutional Review Board at Rhode Island Hospital provided ethical exemption for conducting this study and waived the informed consent requirement for patients whose medical records were analyzed. All research was performed in accordance with DRC government regulations, and no additional local approval was required.

## Results

There were 162 women of child-bearing age with confirmed EVD included for analysis, 27 (16.7%) of whom were pregnant (Fig 1).

Characteristics on admission of pregnant and non-pregnant women are in Table 1. Pregnant Ebola positive women in our study had lower median age (22 [IQR 19, 29] years versus 29 [IQR 21, 27] years for non-pregnant women) and were more likely to report vaginal bleeding and bleeding at an injection site. However, we identified no difference in initial viral load,

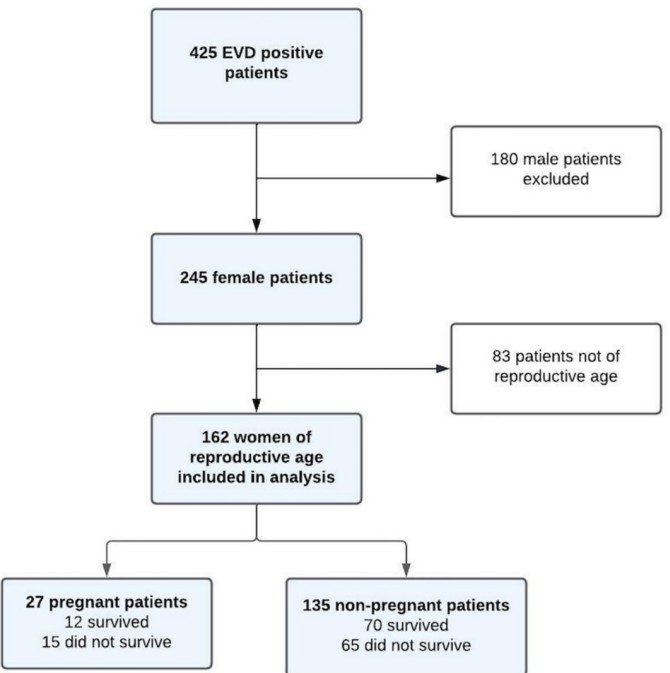

**Fig 1. Flow diagram showing selection of patients included for analysis.**

**Table 1. Characteristics of pregnant and non-pregnant women on admission.**

| Characteristic | Non-Pregnant, N = 135[a] | Pregnant, N = 27[a] | p-value[b] |
|---|---|---|---|
| Demographic Characteristics | | | |
| Age (years), [median, iqr] | 28 (21, 37) | 22 (19, 29) | **0.025** |
| Reported symptoms at triage | | | |
| Days between symptom onset and admission [median, iqr] | 4.0 (2.0, 7.0) | 4.0 (2.0, 6.0) | 0.6 |
| Nausea | 72 (53%) | 9 (33%) | 0.058 |
| Diarrhea | 63 (47%) | 10 (37%) | 0.3 |
| Fatigue | 113 (84%) | 22 (81%) | 0.8 |
| Loss of Appetite | 107 (79%) | 17 (63%) | 0.068 |
| Abdominal Pain | 77 (57%) | 16 (59%) | 0.8 |
| Chest Pain | 44 (33%) | 9 (33%) | >0.9 |
| Bone/Muscle Pain | 68 (50%) | 12 (44%) | 0.6 |
| Joint Pain | 78 (58%) | 16 (59%) | 0.9 |
| Headache | 84 (63%) | 16 (59%) | 0.7 |
| Cough | 40 (30%) | 9 (33%) | 0.7 |
| Breathlessness | 26 (19%) | 4 (15%) | 0.8 |
| Swallowing Problem | 37 (27%) | 5 (19%) | 0.3 |
| Sore Throat | 32 (24%) | 4 (15%) | 0.3 |
| Jaundice | 6 (4.5%) | 1 (3.7%) | >0.9 |
| Conjunctivitis | 71 (53%) | 9 (33%) | 0.062 |
| Rash | 5 (3.7%) | 3 (11%) | 0.13 |
| Hiccups | 2 (1.5%) | 1 (3.7%) | 0.4 |
| Photophobia | 3 (2.2%) | 2 (7.4%) | 0.2 |
| Coma | 5 (3.7%) | 2 (7.4%) | 0.3 |
| Confusion/Disorientation | 4 (3.0%) | 2 (7.4%) | 0.3 |
| Any bleeding | 25 (19%) | 9 (33%) | 0.088 |
| Bleeding from gums | 7 (5.2%) | 2 (7.4%) | 0.6 |
| Bleeding at injection site | 1 (0.7%) | 3 (11%) | **0.015** |
| Bleeding from Nose | 3 (2.2%) | 2 (7.4%) | 0.2 |
| Melena | 9 (6.7%) | 2 (7.4%) | >0.9 |
| Hematemesis | 4 (3.0%) | 1 (3.7%) | >0.9 |
| Coffee ground emesis | 1 (0.7%) | 1 (3.7%) | 0.3 |
| Hemoptysis | 1 (0.7%) | 1 (3.7%) | 0.3 |
| Vaginal bleeding | 6 (4.5%) | 7 (26%) | **0.002** |
| Petechiae/purpura | 0 (0%) | 1 (3.7%) | 0.2 |
| Hematuria | 1 (0.7%) | 1 (3.7%) | 0.3 |
| Other bleeding | 4 (3.0%) | 2 (7.4%) | 0.3 |
| Laboratory characteristics | | | |
| Initial viral load (cycle threshold), [median, iqr] | 21.6 (19.2, 26.2) | 20.5 (17.2, 22.9) | 0.14 |
| Clinical and Epidemiologic Characteristics | | | |
| Vaccinated | 52 (39%) | 11 (41%) | 0.8 |
| Treatment | | | >0.9 |
| Remdesivir | 18 (13%) | 5 (19%) | |
| REGN-EB3 | 46 (34%) | 9 (33%) | |
| MAb114 | 47 (35%) | 9 (33%) | |
| Zmapp | 2 (1.5%) | 0 (0%) | |
| None | 22 (16%) | 4 (15%) | |
| Ebola case contact | 82 (79%) | 14 (67%) | 0.3 |

*(Continued)*

**Table 1.** (Continued)

| Characteristic | Non-Pregnant, N = 135[a] | Pregnant, N = 27[a] | p-value[b] |
|---|---|---|---|
| Outcome | | | |
| Died | 65 (48%) | 15 (56%) | 0.5 |
| Length of stay (days) [median, iqr] | 14 (3, 22) | 6 (2, 24) | 0.8 |

[a]Median (IQR); n (%)

[b]Wilcoxon rank sum test; Pearson's Chi-squared test; Fisher's exact test

vaccination status, or known Ebola case contact. Pregnant and non-pregnant women received similar investigational therapeutics, with most receiving either REGN-EB3 or MAb114. Only two (1.2%) women in our database had a pregnancy test requested during admission, while 45 (27.8%) did not and the remaining 115 (71.0%) had no data available. In crude analysis, pregnant women had similar mortality (56%) compared to non-pregnant women (48%) (chi-square p = 0.5).

We attempted multiple algorithms to obtain balanced comparator groups, including greedy nearest neighbor without replacement and genetic matching. The genetic matching algorithm achieved balance between covariates in the pregnant and non-pregnant women groups with all standardized mean differences (for continuous covariates) and mean differences (for categorical variables) less than 0.1 (Fig 2). Three pregnant patients were excluded by our matching algorithm yielding 24 pregnant patients with 24 matched non-pregnant patients. Additionally, a single non-pregnant patient was excluded prior to completing the matching algorithm because the patient had a missing initial Ct value. The relative risk of death in the DRC cohort among pregnant patients compared to non-pregnant patients was 1.0 (95% CI, 0.58–1.72).

Table 2 lists characteristics and outcomes of the pregnancies identified in the cohort. Eleven (40.7%) pregnancies were first trimester, six (22.2%) were second trimester, and six (22.2%) were third trimester, while four (14.8%) were missing gestational age. Maternal outcomes were similar in first, second, and third trimesters, with maternal death in five (45.4%), three (60.0%), and three (60.0%) mothers in each trimester, respectively. Nine patients (33.3%) received oxytocin and seven (25.9%) received misoprostol.

Three patients (11.1%) were discharged alive with viable pregnancy. One third trimester patient was transferred to another facility for a Caesarian section after active fetal movement

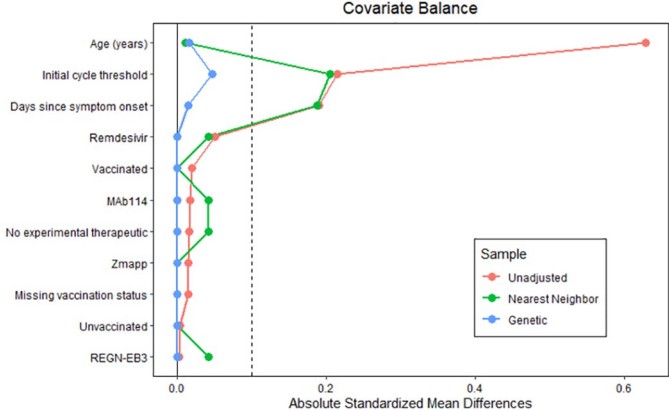

**Fig 2. Covariate balance between in the pregnant and non-pregnant women groups.**

**Table 2. Maternal and fetal outcomes among pregnant patients with EVD grouped by fetal outcome.**

| Trimester | Maternal Outcome | Fetal Outcome | Misoprostol | Oxytocin | Investigational Therapeutic | Vaccinated |
|---|---|---|---|---|---|---|
| **Live Birth** | | | | | | |
| 3 | Death on day 2 from hypovolemic shock, secondary to coagulopathy in setting of EVD and concern for post-partum hemorrhage | Neonate born at term weighing 2.5kg on day 1 of maternal admission. Tested positive for EVD on day 3 of life. Received remdesivir and died in setting of respiratory distress and EVD on day 8 of life. | | X | N/A | |
| 3 | Death soon 14 hours after delivery from likely hypovolemic shock | Infant born alive with APGARs 3, 5, 7 weighing 1500g and admitted to ETC. Received RG on day 1 of life per protocol. Serum PCR negative EVD day 1 of life. Discharged alive on day 10 of life to local nursery and subsequently discharged home. | | | M | X |
| **Discharged alive with viable pregnancy** | | | | | | |
| 3 | Discharged alive, day 20 of admission, | Active fetal movement throughout admission. Transferred for C-section. | | | RG | X |
| 1 | Discharged alive, day 24 of admission | Suspected viable pregnancy through discharge | | | RG | X |
| 1 | Discharged alive, day 29 of admission | Suspected viable pregnancy through discharge | | | M | |
| **Spontaneous abortion** | | | | | | |
| U | Death on day 2 of admission, unclear if related to pregnancy | Spontaneous abortion requiring augmentation | X | | RG | |
| 1 | Discharged alive day 20 of admission | Vaginal bleeding leading to spontaneous abortion requiring augmentation | X | | M | |
| 3 | Discharged alive day 29 of admission | Spontaneous abortion on day 4 of admission requiring augmentation | | X | RG | |
| 2 | Death, day 4 of admission | Developed vaginal bleeding leading to spontaneous abortion requiring augmentation | X | X | RD | X |
| 2 | Discharged alive, day 33 of admission. Noted to have concern for pre-eclampsia while admitted. | Clinician documented loss of fetal movement on day 10 of admission with subsequent delivery | | | M | |
| 1 | Death on day 3 of admission secondary to hypovolemic shock, Ebola complications | Spontaneous abortion requiring augmentation on day 2 of admission | | X | RG | |
| 2 | Death on day 6 of admission | Spontaneous abortion on day 1 of admission with subsequent augmentation | | X | RG | |
| U | Discharged alive day 27 of admission | Spontaneous abortion requiring augmentation for hemorrhage | | X | M | X |
| 1 | Death on day 7 of admission from shock | Spontaneous abortion on day 2 of admission requiring augmentation with further manual curettage on day 5 of admission. | X | X | RG | |
| **Fetal demise/induced delivery** | | | | | | |
| 1 | Discharged alive day 29 of admission | Suspected fetal demise on day 18 of admission with subsequent induction | X | X | M | X |
| 2 | Discharged alive day 20 of admission | Threatened abortion, required digital curettage, oxytocin, misoprostol for hemorrhage | X | X | M | |
| **Maternal death while pregnant** | | | | | | |
| U | Death on day 3 of admission. Noted to have contractions that subsequently led arrest. | Died while pregnant. | | | RD | |
| 3 | Death on day 5 of admission. | Died while pregnant | | | RG | |
| 1 | Death on day 10 of admission | Died while pregnant | | | RD | X |
| 1 | Death on day 4 of admission | Died while pregnant, documented as miscarriage by clinician | | | RD | |
| 3 | Death on day 2 of admission | Died while pregnant | | | N/A | X |
| 2 | Death on day 1 of admission | Died while pregnant | | | N/A | X |

*(Continued)*

**Table 2.** (Continued)

| Trimester | Maternal Outcome | Fetal Outcome | Misoprostol | Oxytocin | Investigational Therapeutic | Vaccinated |
|---|---|---|---|---|---|---|
| 2 | Death on day 3 of admission | Died while pregnant, but noted to have loss of fetal movement prior to maternal death | | | N/A | X |
| 1 | Died on day 3 of admission | Likely died while pregnant | | | M | |
| **Unknown** | | | | | | |
| 1 | Discharged alive on day 46 of admission | Unknown | X | | RD | |
| U | Discharged alive day 33 of admission | Unknown | | | RG | X |
| 1 | Discharged alive day 24 of admission | Unknown | | | M | |

Abbreviations: U–Unknown; M–mAB114; RD–Remdesivir; RG–REGN-113; N/A–Not Available

throughout admission. Two (7.4%) patients were in their first trimester and had minimal documentation of pregnancy during their admission but appeared to be discharged with a viable pregnancy. All three patients received either REGN-EB3 or MAb114, and two were vaccinated.

Eleven (40.7%) patients experienced loss of pregnancy during admission: Nine (33.3%) had spontaneous abortion, and eight (29.6%) required augmentation with misoprostol and/or oxytocin. An additional two patients (7.4%) had suspected fetal demise with subsequent induced delivery. Eight (29.6%) died while pregnant, with two (7.4%) experiencing miscarriage or loss of fetal movement prior to maternal death. Finally, three pregnant patients were discharged alive without documentation of fetal outcomes.

We identified two live births (Table 2) at the ETC, one of whom survived the neonatal period. The surviving infant's mother had been vaccinated against EVD and received Mab114. The infant received REGEN-EB3 monoclonal antibody therapy in the ETC, tested serum negative for EVD on day 1, and was discharged on day 10 of life. Conversely, the non-surviving infant's mother was not vaccinated against EVD and died shortly after arrival in the setting of post-partum hemorrhage and thus was unable to receive an experimental therapeutic. This infant was randomized to receive remdesivir in the ETC, tested positive for EVD on day 3, and died in the setting of respiratory distress on day 8 of life.

## Discussion

Here, we report maternal and fetal outcomes among a cohort of women with EVD during the tenth Ebola outbreak in DRC, and found pregnancy was not associated with increased risk of maternal death [6, 12]. Additionally, we confirmed poor fetal outcomes in women with EVD, with at least 41% of our cohort experiencing loss of pregnancy and one of two live births dying in the first eight days of life [6, 7, 12]. Moreover, both mothers who experienced a live birth died from post-partum hemorrhage.

Our finding that pregnant women were not at increased risk of death compared to propensity score matched non-pregnant women is consistent with reports from the West Africa outbreak as well as a recent systematic review and meta-analysis [12, 26]. While evidence remains limited, the historical assumption that pregnancy is a risk factor for death from EVD is not supported by current evidence. The lack of difference in risk of death between pregnant and non-pregnant may reflect continued high risk of death from EVD in all populations, with death occurring in approximately half of women in each group. Earlier reports may have identified pregnancy as a risk factor for death because of selective reporting of salient cases similar to those we identified, where mothers who gave birth died from post-partum hemorrhage in

the setting of EVD. Regardless, our findings affirm that pregnant women should be eligible for inclusion in all studies for novel vaccines and therapeutics and be eligible for prevention and therapeutics once these are approved.

While perinatal outcomes were poor in our study, our cohort extends the report by Ottoni et al. that improved perinatal outcomes may be possible, particularly considering vaccination and efficacious therapeutics for EVD [13]. We identified an infant born in the ETC discharged alive whose mother was vaccinated and where both mother and infant received a monoclonal antibody. Additionally, we identified a mother with viable pregnancy transferred for Caesarian section who was vaccinated and received a novel therapeutic as well as two other possibly viable first trimester pregnancies.

Our findings raise several concerns including infection control, pregnancy testing, and therapeutics in neonates. First, as fetal and maternal outcomes improve, infection control guidance should be expanded for infants born to mothers with current or previous EVD. Based on our findings, we agree with Ottoni et al. that infants born to mothers with EVD be managed as EVD-suspect and separated from their mothers given that one live birth tested negative for EBOV [13].

Second, we found pregnancy testing was rarely conducted in our cohort. Given that 16.7% of women of child-bearing age were pregnant in our cohort, we suggest women of childbearing age admitted to an ETC be systematically considered for pregnancy testing. Testing would facilitate discussion about risks to the fetus, preparation for spontaneous abortion in patients without clinically obvious pregnancy, and anticipatory guidance regarding infection control at delivery to mothers who survive EVD with viable pregnancy. Likewise, detailed information on pregnancy status, and if pregnant, information about gestational age and fetal outcomes should be collected in future studies.

Third, our findings confirm pregnancy specific management decisions should continue on a case-by-case basis given fetal outcomes remain poor. However, clinicians should be aware that it is possible, though rare, for vaccinated mothers who receive monoclonal antibody therapy to have a viable live birth or be discharged with viable pregnancy.

Fourth, we affirm the standard of care for neonates born in ETCs to mothers with EVD of administering monoclonal antibody therapy regardless of the neonates' EVD test result should continue. While current evidence comes from just three surviving infants, the benefits likely outweigh the risks given neonatal outcomes in the absence of therapeutics are universally poor [13]. The management of neonates born to mothers who survived EVD earlier in pregnancy is less clear. Should they be similarly considered for empiric monoclonal antibody therapy? The possibility of EBOV persistence in the fetus of a mother who received monoclonal antibody therapy is unknown but systematic collection of information on this in the future is essential.

Our study has several strengths, including systematic enrollment of women admitted to the ETC and use of a propensity score model. However, our study has several limitations: First, most pregnancies were identified via self-report or clinician observation so underreporting of pregnancy is possible. Misclassification of pregnant patients as non-pregnant would bias toward finding no mortality difference when one existed. Future systematic pregnancy testing in ETCs would attenuate this concern. Second, because of missing data and small numbers, we were unable to examine whether mothers later in gestation were at higher risk of death from EVD. Finally, our data were collected by clinicians working in challenging circumstances and so assessment of pregnancy outcomes was challenging, with incomplete data on parity status, gestational age, and last menstrual period.

In summary, we report fetal and maternal outcomes among a cohort of pregnant women with EVD in the tenth outbreak of EVD in DRC. We found no evidence of increased mortality secondary to pregnancy but confirmed poor fetal outcomes. Additionally, we identified one

surviving infant born to a mother with EVD and several patients discharged with viable pregnancies.

## Acknowledgments

The authors would like to thank all of the frontline workers and staff at International Medical Corps' Mangina Ebola Treatment Center.

## Author Contributions

**Conceptualization:** David Philpott, Neil Rupani, Monique Gainey, Prince Imani Musimwa, Shiromi M. Perera, Razia Laghari, Mija Ververs, Adam C. Levine.

**Data curation:** David Philpott, Neil Rupani, Shiromi M. Perera, Razia Laghari, Mija Ververs, Adam C. Levine.

**Formal analysis:** David Philpott, Neil Rupani, Eta N. Mbong, Prince Imani Musimwa, Shiromi M. Perera, Mija Ververs, Adam C. Levine.

**Investigation:** David Philpott, Monique Gainey, Eta N. Mbong, Prince Imani Musimwa, Mija Ververs.

**Methodology:** David Philpott, Neil Rupani, Shiromi M. Perera, Mija Ververs.

**Project administration:** Monique Gainey.

**Software:** David Philpott.

**Supervision:** David Philpott, Neil Rupani, Mija Ververs, Adam C. Levine.

**Validation:** David Philpott, Neil Rupani.

**Writing – original draft:** David Philpott, Neil Rupani.

**Writing – review & editing:** David Philpott, Neil Rupani, Monique Gainey, Eta N. Mbong, Prince Imani Musimwa, Shiromi M. Perera, Adam C. Levine.

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
