## [Decision Letter · Decision Letter 0]

27 Jan 2023

PONE-D-22-28805Maternal, fetal, and perinatal outcomes among pregnant women admitted to an Ebola treatment center in the Democratic Republic of Congo, 2018–2020PLOS ONE

Dear Dr. Philpott,

Thank you for submitting your manuscript to PLOS ONE. After careful consideration, we feel that it has merit but does not fully meet PLOS ONE’s publication criteria as it currently stands. Therefore, we invite you to submit a revised version of the manuscript that addresses the points raised during the review process.

Please include the following items when submitting your revised manuscript:A rebuttal letter that responds to each point raised by the academic editor and reviewer(s). You should upload this letter as a separate file labeled 'Response to Reviewers'.A marked-up copy of your manuscript that highlights changes made to the original version. You should upload this as a separate file labeled 'Revised Manuscript with Track Changes'.An unmarked version of your revised paper without tracked changes. You should upload this as a separate file labeled 'Manuscript'.If applicable, we recommend that you deposit your laboratory protocols in protocols.io to enhance the reproducibility of your results. Protocols.io assigns your protocol its own identifier (DOI) so that it can be cited independently in the future. For instructions see: https://journals.plos.org/plosone/s/submission-guidelines#loc-laboratory-protocols. Additionally, PLOS ONE offers an option for publishing peer-reviewed Lab Protocol articles, which describe protocols hosted on protocols.io. Read more information on sharing protocols at https://plos.org/protocols?utm_medium=editorial-email&utm_source=authorletters&utm_campaign=protocols.

We look forward to receiving your revised manuscript.

Kind regards,

Pierre Roques, Ph.D.

Academic Editor

PLOS ONE

Journal Requirements:

2. During our internal checks, the in-house editorial staff noted that the original data was collected in another country. Please check the relevant national regulations and laws applying to foreign researchers and state whether you obtained the required permits and approvals. Please address this in your ethics statement in both the manuscript and submission information. In addition, please ensure that you have suitably acknowledged the contributions of any local collaborators involved in this work in your authorship list and/or Acknowledgements. Authorship criteria is based on the International Committee of Medical Journal Editors (ICMJE) Uniform Requirements for Manuscripts Submitted to Biomedical Journals - for further information please see here: https://journals.plos.org/plosone/s/authorship.

Reviewers' comments:

Reviewer's Responses to Questions

**Comments to the Author**

1. Is the manuscript technically sound, and do the data support the conclusions?

Reviewer #1: Yes

2. Has the statistical analysis been performed appropriately and rigorously? 

Reviewer #1: Yes

3. Have the authors made all data underlying the findings in their manuscript fully available?

Reviewer #1: No

4. Is the manuscript presented in an intelligible fashion and written in standard English?

Reviewer #1: Yes

5. Review Comments to the Author

Reviewer #1: Overall this study by Philpott and colleagues adds to the growing body of knowledge on EVD in pregnancy. the manuscript is well written and the results well presented.

A general comment would be to read carefully through and correct typographical errors prior to resubmission/publication.

Introduction and rationale: this was well written and provided enough background information justifying the study

Methods: the methods are succinct and well described

Results: the results are clear and presented in full

Discussion and conclusions: while the discussion supports the results I would like the authors to elaborate a little more on possible reasons for the mortality relative risk of 1.0. the study objectives indicate you wish to compare mortality in pregnant and non-pregnant women and while your findings are interesting and supported both by a recent systematic review Kayem et al 2022 and some studies from the West African Ebola outbreak, you donot highlight these and right now your discussion on this finding is a single sentence at the end of your discussion section. I would like you to give us your opinion on possible reasons why no difference in mortality is found.

Additionally based on the findings in your study, what are your recommendations for practice, research and policy.

Data availability: Data not provided because authors indicate restrictions apply. they would need to specify the restrictions.

6. PLOS authors have the option to publish the peer review history of their article (what does this mean?). If published, this will include your full peer review and any attached files.

Reviewer #1: No

---

## [Author Response · Author response to Decision Letter 0]

17 Apr 2023

Introduction and rationale: this was well written and provided enough background information justifying the study.

Thank you.

Methods: the methods are succinct and well described.

Thank you.

Results: the results are clear and presented in full.

Thank you.

Discussion and conclusions: while the discussion supports the results, I would like the authors to elaborate a little more on possible reasons for the mortality relative risk of 1.0. The study objectives indicate you wish to compare mortality in pregnant and non-pregnant women and while your findings are interesting and supported both by a recent systematic review Kayem et al 2022 and some studies from the West African Ebola outbreak, you do not highlight these and right now your discussion on this finding is a single sentence at the end of your discussion section. I would like you to give us your opinion on possible reasons why no difference in mortality is found.

Additionally based on the findings in your study, what are your recommendations for practice, research and policy.

Thank you for your requested clarification. We have expanded our discussion with additional consideration for why we did not identify an increased risk of death among pregnant patients in the study and included a reference to the Kayem et al review as well as data from the West Africa Outbreak. Likewise, we have expanded recommendations for future data collection to include pregnant women as an important subgroup.

---

## [Decision Letter · Decision Letter 1]

25 May 2023

Maternal, fetal, and perinatal outcomes among pregnant women admitted to an Ebola treatment center in the Democratic Republic of Congo, 2018–2020

PONE-D-22-28805R1

Dear Dr. Philpott,

We’re pleased to inform you that your manuscript has been judged scientifically suitable for publication and will be formally accepted for publication once it meets all outstanding technical requirements.

Kind regards,

Pierre Roques, Ph.D.

Academic Editor

PLOS ONE

Additional Editor Comments (optional):

Reviewers' comments:

Reviewer's Responses to Questions

**Comments to the Author**

1. If the authors have adequately addressed your comments raised in a previous round of review and you feel that this manuscript is now acceptable for publication, you may indicate that here to bypass the “Comments to the Author” section, enter your conflict of interest statement in the “Confidential to Editor” section, and submit your "Accept" recommendation.

Reviewer #1: All comments have been addressed

2. Is the manuscript technically sound, and do the data support the conclusions?

Reviewer #1: Yes

3. Has the statistical analysis been performed appropriately and rigorously? 

Reviewer #1: Yes

4. Have the authors made all data underlying the findings in their manuscript fully available?

Reviewer #1: No

5. Is the manuscript presented in an intelligible fashion and written in standard English?

Reviewer #1: Yes

6. Review Comments to the Author

Reviewer #1: All my comments have been appropriately addressed and I recommend that the manuscript is accepted for publication

7. PLOS authors have the option to publish the peer review history of their article (what does this mean?). If published, this will include your full peer review and any attached files.

Reviewer #1: No

---

## [Editor Report · Acceptance letter]

29 Aug 2023

PONE-D-22-28805R1 

Maternal, fetal, and perinatal outcomes among pregnant women admitted to an Ebola treatment center in the Democratic Republic of Congo, 2018–2020 

Dear Dr. Philpott:

I'm pleased to inform you that your manuscript has been deemed suitable for publication in PLOS ONE. Congratulations! Your manuscript is now with our production department. 

Kind regards, 

on behalf of

Dr. Pierre Roques 

Academic Editor

PLOS ONE